# Associations between Home Environment, Children’s and Parents’ Characteristics and Children’s TV Screen Time Behavior

**DOI:** 10.3390/ijerph18041589

**Published:** 2021-02-08

**Authors:** Carolina Bassul, Clare A. Corish, John M. Kearney

**Affiliations:** 1School of Biological and Health Sciences, Technological University Dublin, City Campus, Kevin Street, Dublin 8, Ireland; 2School of Public Health, Physiotherapy and Sports Science, University College Dublin, Belfield, Dublin 4, Ireland; clare.corish@ucd.ie

**Keywords:** pre-school children, screen time, home environment, parental role modeling

## Abstract

In Ireland, television (TV) screen time is a highly prevalent sedentary behavior among children aged less than five years. Little is known about the influence of parental rules and policies or screen time availability and accessibility within the home on children’s TV screen time behaviors. This cross-sectional study aimed to examine the extent to which parents’ sociodemographic and sedentary behaviors are associated with children’s TV screen time; and to determine the associations between parents’ rules and practices, home physical environment and children’s daily TV viewing. Three hundred and thirty-two children aged 3–5 years and their parents participated in the study. Children’s TV screen time and home environmental characteristics (parents’ rules and practices and the physical environment) were assessed using questions from standardized and validated questionnaires. The data were analyzed using binary logistic regression. Within the different sedentary behaviors evaluated, parents’ TV viewing was positively associated with children’s TV screen time (OR 1.65, 95%CI 1.09–2.50, *p* = 0.018). Leaving the TV on, whether or not it was being watched, was associated with a 38% increased probability of children watching ≥ 1 h TV daily. Children whose parents restricted their outdoor activity were more likely to watch ≥ 1 h TV daily (OR 2.01, 95%CI 1.04–3.88, *p* = 0.036). Findings from the study demonstrated that parents’ own screen time behaviors, leaving the TV on whether it was being watched or not and restricting outdoor play were associated with higher children’s TV viewing in the home environment. This knowledge is essential to inform future interventions aimed to address the increase in screen time among young children.

## 1. Introduction

Screen-based entertainment such as watching television (TV), playing video games and using smartphones and tablets [1] has been shown to be highly prevalent among young children [2]. TV screen time is the most common sedentary behavior among children aged under five years [3]. Excessive screen time (≥1 h daily) [4] in early childhood may be associated with aggressive and anti-social behaviors, sleep pattern disturbances, poor motor development and overweight/obesity [1,5].

Recommendations to limit screen-related behavior in young children are in place in many countries including Australia [6], Canada [7], the United States of America (USA) [8] and New Zealand [9]. The Irish National Guidelines follow the recent WHO recommendations for screen time behavior for children under 5 years; for example, 3–4-year-old children should not spend more than 1 hour on screen-related sedentary behavior [1,10]. Conversely, the United Kingdom’s (UK) most recent screen time guideline developed by the Royal College of Pediatrics and Child Health (RCPCH) did not establish a specific cut-off for children’s screen time due to weak evidence of the association between screen time and poor health outcomes [11]. The UK guidelines highlight that screen time should not interrupt positive activities for children such as socializing, exercise and sleep and that families should negotiate screen time limits with their children [11]. Screen time reported by parents in Ireland indicated that pre-school children watch an average of 2.2 h of TV daily, which is more than double the WHO recommendation [12,13]. A similar trend has been observed in other countries; for example, in the UK, 96% of 3–4-year-old children watch an average of 2 h TV daily [14], in the USA, children watch TV for an average of 2.2 h [8] while in Australia, pre-school children watch TV for over 1.5 h daily [15].

Within the home environment, parents play a pivotal role in pre-school children’s screen time sedentary behavior. Parents’ sociodemographic and behavioral characteristics, such as education level and role modeling for sedentary behavior, are associated with children’s screen time [16,17,18]. For example, children whose parents had a lower level of education had more overall screen time at home compared with children whose parents had a higher education level [19]. A systematic review demonstrated evidence of parents’ own sedentary behavior being linked to children’s screen time. Nine out of ten studies reported a positive association between parents’ TV watching and children’s screen time [20].

Previous studies showed that parents’ practices around children’s screen time is also associated with children’s TV viewing [21]. Such practices refer to parents’ behaviors and interactions which may restrict or encourage children’s use of electronic media. A European cross-sectional study, which included five countries (Belgium, Germany, Greece, Hungary and Norway) and 3325 child/parent dyads, reported that the presence of rules on how much time children were allowed to watch TV or use computers/game consoles was associated with children’s decreased screen time behavior [21]. Results from another cross-sectional study showed that parenting practices of both mothers and fathers have a similar influence on the amount of screen time in 1.5–5-year-old children. During weekdays, both parents’ practice of having screen time during mealtimes was positively associated with children’s screen time, while monitoring and limiting screen time were inversely associated with children’s media use. For weekends, using screen time to control behavior (e.g., reduced TV if child misbehaves) was positively associated with children’s screen time [22]. 

In addition, practices which encourage physical activity and active play in the home have also been found to be inversely associated with children’s screen-related sedentary behavior [21]. One recent study evaluated the link between parenting practices that can limit or encourage physical activity and pre-school children’s preference for screen time sedentary behaviors. The authors reported that pre-school children whose parents restricted outdoor play tended to prefer sedentary activity over physical activity [23]. 

A number of systematic reviews have similarly reported associations between the home physical environment and screen-based sedentary behaviors [24,25,26]. For instance, one review [24] reported that the availability of media equipment at home and children’s screen time were positively associated in 10 of 16 studies. This review also indicated that availability of media in children’s bedrooms and children’s sedentary behaviors were positively correlated in 50% of the included studies. On the other hand, the associations between media equipment generally available in the home and physical activity were inconsistent [24]. The availability of physical activity equipment, such as a bicycle, sports equipment, jump rope or active video game, was unrelated to moderate–vigorous physical activity in young children but was inversely associated with children’s sedentary behaviors. The authors hypothesized that having physical activity equipment at home may decrease sedentary behaviors by increasing light physical activity instead of moderate–vigorous activity [24]. 

Understanding the characteristics of the home environment associated with pre-school children’s TV screen time is essential for the promotion of children’s health in a highly technological world and to provide parents with information on the appropriate use of screen time [27]. In Ireland, little is known about the influence of the home environment, including parents’ rules and policies or screen time availability and accessibility in the home, on children’s TV screen time behaviors. The results from our previous study showed a high prevalence of TV viewing among pre-school children and that this behavior was independently associated with markers of unhealthy dietary intake [28]. Therefore, the purpose of undertaking the present study was to examine the extent to which parents’ sociodemographic and sedentary behaviors are associated with children’s TV screen time, and to determine the associations between parents’ rules and practices, the home physical environment and children’s daily TV viewing. As such, understanding parents’ behaviors and home environment characteristics around children’s screen viewing is essential to identify potential areas for intervention to decrease home screen time, increase physical activity and consequently overall health in this population group [29].

## 2. Materials and Methods 

### 2.1. Participants and Study Procedures

Data collection started in August 2016 and was completed in June 2017 with data collected in 2–3 pre-schools per month. Children and their parents/guardians (hereon referred to as “parents”) were recruited from 25 pre-schools located across Dublin. A total of 670 children aged 3–5 years, free from any medical condition which affects growth and development, and their parents were eligible to participate in the present study. Parents with insufficient English proficiency to complete the questionnaire were excluded. In each pre-school, parents were invited to participate in the study by the pre-school manager using a “pack” which contained the study information letter, consent forms (study consent form and children’s anthropometric measurement consent form) and a questionnaire (the “Pre-schoolers Health Study” questionnaire) to be completed by one parent of each child. Parents were given one week to complete the questionnaire and return it to the pre-school. A reminder was sent by the pre-school’s staff to parents who failed to return the completed questionnaire and one more week was given [28]. The study was approved by the Ethics Committee of the Technological University Dublin (TUD) (Ref 15–109).

### 2.2. Instrument of Data Collection: “Pre-Schoolers Health Study” Questionnaire 

#### 2.2.1. Characteristics of Participants 

The general information section of the “Pre-Schoolers Health Study” questionnaire assessed some of the children’s and parents’ characteristics such as children’s date of birth, gender and time spent in pre-school (e.g., “full-time” or “part-time”). Parent’s age, self-reported weight and height, education level, nationality and marital status were also reported [21]. 

Children’s anthropometric data were collected by the researcher (CB) at each pre-school with the assistance of trained pre-school staff. Weight was obtained using calibrated electronic scales, Seca model 899, which measures children to the nearest 0.1 kg. Stature was obtained using a stable, calibrated stadiometer, the Leicester Height Measure; this measures height to the nearest 0.5 cm. Children’s heights and weights were converted to z-BMI-for-age using the WHO Anthro software calculator [30].

##### Children’s TV Screen Time

To assess children’s TV screen time, parents responded to the question “On average how many hours per day does your child watch any type of TV including DVDs and videos?”. Response options were 1 = none; 2 = less than 30 min a day; 3 = 30 min to 1 hour a day; 4 = 1 to 1.5 h a day; 5 = 1.5 to 2 h a day; 6 = more than 2 h a day [21]. The responses were then dichotomized as (0) < 1 h daily or (1) ≥ 1 h daily, in accordance with recent WHO recommendations for maximum screen time in children aged < 5 years old [1].

##### Parents’ Physical Activity and Sedentary Behaviors

Parents’ moderate and vigorous physical activity levels were assessed separately through the question: “How many days of the week do you engage in moderate exercise for at least 30 min? (moderate exercise includes brisk walking, cycling slower, general garden, tennis (double) medium, paced swimming), or how many days of the week do you engage in vigorous exercise for at least 30 min? (vigorous exercises includes jogging or run, sports such as football, squash and aerobics, or fast cycling or heavy gardening)”. Possible responses ranged from 1–5 (1 = less than once a week; 2 = 1 to 2 times a week; 3 = 3 to 4 times a week; 4 = 5 or more times a week; 5 = never). These were then re-coded as a single variable according to the National Physical Activity Plan for Ireland as ≥ 5 times per week, 1–4 times per week, or None [22]. To assess their sedentary behavior, parents provided answers to the statement: “On a typical weekday, how much time do you spend (from when you wake up until you go to bed) doing the following? Watching any type of TV including DVDs and videos; playing video games such as PlayStation, Nintendo, i-PAD and computer games; sitting reading a book or magazine, listening to music, doing art or craft work; sitting doing paperwork or computer work; travelling by car/public transport”. A 6-point Likert scale (1 = none, 2 = 30 min or less, 3 = 1 to 2 h, 4 = 2 to 3 h, 5 = 3 to 4 h, 6 = 5 h or more) was used to assess parents’ sedentary behaviors. The weekday and weekend item scores were summed giving a continuous variable (2 to 12) with high scores representing more time spent in sedentary behavior.

#### 2.2.2. Parents’ Rules and Practices around Children’s TV Screen Time

Variables assessing parents’ rules around children’s TV screen time were adapted from the Healthy Home Survey (HHS) [31]: (i) parents set rules around children’s screen time; (ii) parents used screen time as reward and punishment (parents responded as “yes” or “no” to these questions); (iii) parents allowed their children to eat meals or snacks while watching TV. Parents responded on a 4-point Likert scale (1 = frequently, 2 = sometimes, 3 = occasionally and 4 = rarely/never). A dichotomization was generated by collapsing responses 1 and 2 to “frequently” and responses 3 and 4 to “rarely/never”. In addition, parents’ practice of leaving the TV turned on, whether it is being watched or not, and parents allowing their children to play inside and outside actively were also assessed using a 4-point Likert scale (1 = rarely/never; 2 = sometimes; 3 = most of the time; 4 = all the time). These were then dichotomized by collapsing responses 1 and 2 to “not all the time” and responses 3 and 4 to “all the time”.

#### 2.2.3. Home Physical Environment

Home physical environment variables were assessed using questions from the Healthy Home Survey (HHS) [31] and the Physical and Nutritional Home Environment Inventory (PNHEI) [32] questionnaires. The home physical environment variables were: (i) parents’ perceptions of the size of their outdoor garden and space (small, medium or large); (ii) usable play equipment such as trampolines, swings and slides, bicycles, tricycle and scooters, (iii) number of televisions in the home; (iv) presence of cable or satellite TV; (v) presence of TV in a room where meals are eaten; (vi) presence of TV or game consoles in the child’s bedroom and the child’s access to game consoles. Parents responded “yes” or “no” to these questions.

### 2.3. Statistical Analysis

Statistical analyses were carried out using IBM SPSS for macOS Mojave, version 26.0 (IBM, New York, NY, USA). Bivariate analyses were first performed on the characteristics of the home environment (parent and children characteristics, parents’ rules and practice and home physical environment) and children’s TV screen time. The associations between categorical variables were assessed using cross-tabulations, and Chi-squared tests were used to assess statistical significance. The association between continuous normally distributed variables and categorical variables (children’s TV screen time) were assessed using independent sample t-tests, while the association between non-normally distributed variables and children’s TV screen time were assessed using Mann–Whitney U tests.

Adjusted binary logistic regression was then performed on characteristics of the home environment associated with children’s TV screen time in the bivariate analysis. The Forced Entry Method was used, whereby all independent variables were tested in one block to assess their association while controlling for the effects of other variables in the model [33]. The multicollinearity was checked by running collinearity in the multiple linear logistic regression and was measured by tolerance < 0.10 and a variance inflation factor (VIF) > 10. There was no evidence of collinearity in the adjusted model. The statistical significance level was set at *p* < 0.05. The adjusted exponentiation of the B coefficient-Exp (B) value or odds ratios (OR) and 95% confidence interval (CI) were reported for each independent variable [28].

## 3. Results

### 3.1. Characteristics of Participants

In total, 332 (49.5% of those eligible) children/parent dyads participated in the present study. The descriptive characteristics of participants are presented in Table 1. The mean age of the children included in this study was 4.37 (SD 0.51), with 50.2%, *n =* 165, boys and 49.8%, *n* = 164, girls, and 63.6%, *n* = 210, of children attended pre-school part-time (<5 h daily). The majority of parents were: within the age range of 30–39 years (63.6%, *n* = 211), mothers (88.3%, *n* = 293), highly educated (56.6%, *n* = 188), married or living with their partner (81.3%, *n* = 270) and of Irish nationality (68.6%, *n* = 223). The descriptive analyses of parents’ physical activity showed that for over half of the parents, the level of physical activity did not meet the Irish guideline of 30 minutes of moderate–vigorous exercise five times per week. Regarding sedentary activities, parents tended to watch more TV (median 6.0, IQR 5.0–7.0) than sitting reading (median 5, IQR 4.0–6.0) and sitting doing paperwork or computer work (median 5, IQR 3.0–7.0).

### 3.2. Characteristics of the Home Environment

The characteristics of the home environment are outlined in Table 2. The majority of parents reported that they set rules on children’s TV viewing (79.2%, *n =* 283) and tended to reduce the TV time if the child misbehaved (72%, *n* = 239). Leaving the TV on whether it was being watched or not was a common practice for 21.1%, *n* = 70, of parents. Active play rules were frequently reported by parents, for example, children were allowed to play actively inside “all the time” (64.2%, *n* = 212) but not allowed to play outdoors “all the time” (63.6%%, *n* = 221). Almost all households had an outdoor garden or space (91%, *n* = 300) with play equipment such as trampolines, slides and swings available in 48.9%. The majority of households (85.2%, *n* = 281) had at least one television with cable or satellite TV. Only a few children had a TV or game console in their bedroom (13.3*%, n =* 44 and 5.7%, *n* = 19, respectively).

### 3.3. Characteristics of the Partcipants and Home Environment Associated with Children’s TV Screen Time

Table 3 shows the participant and home environmental characteristics associated with children’s TV screen time after bivariate analysis. From thirty-five variables tested, twelve showed associations with children’s TV screen time behaviour: Children attending pre-school full time/part time, parents’ education level and parents’ TV viewing, parents setting rules around children’s TV and game console, children has access to game console, parents allowing their child to eat meals and snacks in front of the TV, parents allowing their children to play outside, leave the TV on whether or not it is being watched, having TV in a room where meals are eaten, having cable and satellite TV, and the number of TVs in the house.

The adjusted model resulting from the analyses of all home environmental characteristic variables associated with children’s TV screen time are presented in Table 4. In total, six of twelve independent variables made a statistically significant contribution to the adjusted model. Attending pre-school part-time was positively correlated with children’s daily TV viewing, increasing by 18% the likelihood of watching ≥ 1 h of TV daily compared to children who attended pre-school full-time (OR 1.18, 95%CI 1.03–3.18, *p* = 0.039). Children whose parents spent greater time watching TV were 65% more likely to watch ≥ 1 h TV daily compared to children whose parents spent less time watching TV daily (OR 1.65, 95%CI 1.09–2.50, *p* = 0.018). Furthermore, having a TV in the same room where meals are eaten and eating snacks while watching TV increased the likelihood of children watching ≥ 1 h TV daily by over 2 and 2.5 times, respectively. Children whose parents restricted their outdoor play activity were less likely to comply with the WHO recommendations for maximum children’s screen time (OR 2.01, 95%CI 1.04–3.88, *p* = 0.036). Having the TV on most of the time whether or not it was being watched was associated with 38% increased probability of children watching ≥ 1 h TV daily.

## 4. Discussion

Within the home environment, children’s TV screen time was shown to be consistently associated with children attending pre-school part-time, their parents’ screen-related sedentary behaviors, parents’ rules and practices around their children’s screen time and some aspects of the home physical environment such as leaving the TV on whether or not it is being watched. In the present study, children who attended pre-school less than 5 h a day were more likely to watch more TV daily compared to children who attended pre-school full-time. Indeed, a cross-sectional study with 149 pre-schoolers reported that children who attended daycare for more than 4 hours daily were less likely to engage in screen time (television and game console) compared to children who were minded at home by parents [34]. Furthermore, a systematic review of the literature showed that children in center-based childcare were less exposed to screen time compared to children in home-based childcare [35]. However, such results must be interpreted with caution as parents were unable to report children’s screen time while in pre-school care. In addition, the present study did not assess the presence of screen time policies and children’s media accessibility in pre-school settings which is an important factor to examine in order to understand children’s screen time behavior within childcare centers [35,36].

Our results also showed that those parents who watched more TV daily were more likely to have children who watched a greater amount of TV. This clearly highlights the influence of role modeling on children’s screen-related behaviors. Parents who exceed screen time recommendations tend to be more likely to prioritize screen time as a shared family activity and have a more relaxed approach to managing their children’s screen time behavior [37]. In contrast, parents who set limits around their children’s TV time tended to create a home environment supportive of physical activity, and such children tended to spend more time engaging in active play [5,23,37,38]. In evaluating the relationship between parent–child screen time behaviors [23], it is also important to consider parenting practices such as encouragement and support for physical activity [39], parents’ perceptions about screen time (e.g., “too much screen time”), and parents’ self-efficacy (e.g., having the confidence to say “No” when children ask for screen time [20].

Parents allowing children to eat snacks in front of the TV and the presence of TV in the same room where meals are eaten were also positively associated with higher daily TV viewing in children. Our findings concur with the results from a cross-sectional study with 847 children aged 2 to 5 years old, which reported that children’s screen time during mealtimes was positively associated with children’s overall screen time and TV exposure [40]. Such a relationship might be explained by the fact that parents who allowed children to watch more TV daily might also be more inclined to have fewer rules around the screen time environment and allow their children to eat in front of the TV [41]. We hypothesized that in households where there is a TV in the same room where meals are eaten, parents might be less restrictive with children’s TV screen time during mealtimes and that they have a regular habit of eating while watching TV [42].

Home media practices such as leaving the TV turned on, whether it is being watched or not, have been linked to children having lower attention during playtime and less well developed cognitive skills, as well as poorer quality parent–child interaction [43]. A large American study with 1452 children aged between 8 months and 8 years demonstrated that, on average, children were exposed to 3.87 h of background TV daily. Infants/toddlers were the most exposed to background TV, with 5.5 h exposure on a typical day [43]. In the present study, background TV exposure was positively associated with higher children’s TV viewing. Indeed, a study with 1051 parents of children aged from 6 months to 6 years reported that children who live in a household where the TV was on all or most of the time watched an average of 25 min more television per day, compared to children who lived in a household where the television is on half of the time or less [42]. Background television is mostly likely to occur in a household where parents have a positive attitude towards media and, therefore, tend to set fewer rules around children’s screen time [42].

A home physical environment more supportive of physical activity has been shown to be negatively associated with children’s screen time [44]. For the pre-school age, previous studies have demonstrated that outdoor play is consistently associated with higher physical activity levels, and inversely associated with sedentary time [23,45,46]. In the present study, the majority of parents reported aspects of the home physical environment supportive of outdoor play and physical activity such as the presence of an outdoor garden and space with usable play equipment (e.g., trampoline, slide, bicycle and scooter). Importantly, parental restriction of outdoor play appeared to be associated with higher TV viewing in their children. Similarly, a recent cross-sectional study showed that children whose parents applied restrictive rules around outdoor play were more likely to engage in screen time activities. The authors suggested that parental restrictive rules around outdoor play may reflect parents’ busy schedules and safety concerns; in this context, the screen time may be used as a safe “babysitter” option [23]. In the present study, weather may be a limiting factor for children’s outdoor play and may consequently impact on children’s TV viewing. Since Ireland is well known for its abundant rainfall [47], parents may see the inclement weather as a barrier to children’s outdoor play. Perhaps to overcome such a barrier, parents should be provided with specific guidance on the importance of outdoor play and appropriate dress for wet and/or cold weather. In another study involving toddlers, an association between the presence of active play equipment and reduced screen time was found. In summary, the presence of such equipment at home might encourage children to engage in more active play and fewer screen-based activities [15].

## 5. Strengths and Limitations

To our knowledge, this is the first study examining the associations between the home environment and pre-school children’s TV screen time in Ireland. As home environmental characteristics and children’s sedentary screen-time-related behaviors have been found to be related to socioeconomic status [48,49,50], an important strength of this study is the representation (over 45%) of parents of low education level, an indicator of socioeconomic status [51]. A further strength of our study is the use of standardized and validated instruments, which allows for direct comparison with previous studies in the same age group.

This study also has its limitations. The data collected were not nationally representative. The recruitment of child–parent dyads in pre-school and school settings has been previously acknowledged as challenging in a number of studies [52,53,54,55] with difficulties experienced in achieving the high participation rates important to ensure representativeness of research findings [53]. In the present study, strategies to increase and overcome such challenges included: provision of an information leaflet to all parents that described the study prior to starting the study; colorful posters in the participating pre-schools; provision of detailed information about the study to participating parents with all information materials written in an accessible style to ensure clarity and comprehension; multiple contacts with parents and pre-schools; personalized research questionnaire packs that included the university logo; provision of a pen within the questionnaire packs; and complete assurance about the anonymity of the study. In addition, the field researcher (CB) ensured a friendly and ongoing relationship with pre-school managers and staff [53]. Despite these measures, the sample representativeness was only 49.5% which is below the 70% needed to moderate the negative effects of non-response bias [56,57]. Therefore, this may be a limiting factor and the results must be interpreted with caution and may not be generalizable. Our child/parent dyads were recruited through a randomized stratified sampling of pre-schools in different geo-socioeconomic areas of Dublin, Ireland. Random sampling is a simplified sampling approach, and it comes with a risk of sampling error which can also potentially lead to bias in the results [56]. Furthermore, the present study sample size can be explained by the full research project framework, which evaluated different environmental elements associated with pre-school children’s health, nutrition and physical activity/sedentary behaviors. For example, the sample recruitment and data collection were carried out by one researcher who, in addition to gathering quantitative and qualitative data on the parent/child dyads, also collected information on each pre-school’s nutrition and physical activity through observation, as well as data on the neighborhood environment.

Only 9.9% (*n* = 33) of the participants were fathers. Assessing fathers’ influence on children’s TV screen time would provide a relevant insight into this behavior in the home environment [58]. Many sedentary behaviors such as time spent on other types of media such as iPads, computers, electronic games, and reading, drawing and quiet play are difficult to measure in the pre-school population. Measurement relies basically on parental proxy-reports as most pre-school children do not have the cognitive ability to self-report. Therefore, TV viewing was the only sedentary behavior included in this analysis. Children’s TV screen time over the weekend, or while in formal (at home with childminder) or informal care, was not explicitly captured by the questionnaire. Furthermore, parents’ ability to accurately report the intensity of their own physical activity may be a limiting factor. The self-completed questionnaire used in the present study may have induced positive response bias; therefore, the results must also be interpreted with caution. The study design is cross-sectional; therefore, it can only demonstrate associations between the variables investigated but cannot demonstrate cause and effect.

## 6. Conclusions

With technology and screen time behaviors being part of modern life, such behaviors in children pose a growing challenge. The holistic approach taken in this study provides a contribution to the literature on various home environmental characteristics and their association with children’s TV screen time. For example, findings from the present study identified that parents’ behaviors and practice were associated with their children’s TV screen time in the home environment. Parents own TV viewing and rules on outdoor active play were positively associated with children’s TV viewing behavior. Based on the present findings, guidelines should provide more than a simple cut-off to limit screen time. Parents need tools and information on how to set rules for screen time, how to introduce non-screen related activity into children’s daily routines and how to control their own screen time behaviors. The study findings also showed an association between children’s TV exposure, presence of TV in the same room where meals are eaten and parents allowing their children to eat while watching TV and higher children’s TV screen time. Given the negative short- and long-term health outcomes of excessive screen time and eating while watching television, interventions to decrease this habit in both children and their parents could prove beneficial. In addition, campaigns and programs that incentivize families to spend more time in interactive activities at home or outside and, consequently, limit family and children’s screen time may also be helpful.

## Figures and Tables

**Table 1 ijerph-18-01589-t001:** Characteristics of participants.

Characteristic	*n*	%	Mean	SD
Children				
Age (*n* = 332)			4.37	0.51
Gender (*n* = 329)				
Female	164	49.8		
Male	165	50.2		
Child z-BMI (*n* = 204)			0.74	0.92
Attending pre-school (*n =* 330)				
Full-time	120	36.4		
Part-time	210	63.6		
Children’s TV screen time				
≥1 h daily	186	56.0		
<1 h daily	146	44.0		
Parents				
Age (*n* = 332)				
20–29 years	40	12		
30–39 years	211	63.6		
40 or more years	81	24.4		
Relationship with child (*n =* 332)				
Mother	293	88.3		
Father	33	9.9		
Other	6	1.8		
Nationality (*n* = 325)				
Irish	223	68.6		
Eastern European	45	13.6		
Western European	24	7.2		
Asian	16	4.8		
African	8	2.4		
South American	5	1.5		
Other	4	1.2		
Marital status (*n* = 332)				
Married/living together	270	81.3		
Education level (*n* = 332)				
Undergraduate and post-graduate education level	188	56.6		
Secondary school or less education level	144	43.4		
Number of children in the household				
1 child	84	25.3			
2 children	141	42.5			
≥3 children	107	32.2			
Household income (*n* = 263)				
<40.000 €/p.a.	86	32.7		
≥40.000 €/p.a.	177	67.3		
Parents’ moderate to vigorous exercise (*n =* 326)				
≥5 times per week	104	31.9		
1–4 times per week	172	52.8		
None	50	15.3		
Parents’ sedentary behavior (range 2 to 12)			median	IQR
Watching TV (n = 329)			6.0	5.0–7.0
Playing video games (n = 329)			2.0	2.0–2.0
Sitting reading or listening to music (n = 329)			4.0	4.0–6.0
Sitting doing paperwork or computer work (*n* = 329)			5.0	3.0–7.0

BMI: body mass index, IQR: interquartile range; TV: television, SD: standard deviation; p.a.: per annum. Source: Bassul, C.; Corish, C.A.; M. Kearney, J (2020) [28].

**Table 2 ijerph-18-01589-t002:** Home environmental characteristics: Parents’ rules and practices and home physical environment.

Characteristic	*n*	%	Median	IQR
Parents’ rules and practice around children’s TV screen time				
Set rules around TV and game console (*n =* 330)	263	79.7		
Reward good behavior with TV (*n* = 329)	143	43.5		
Reduce TV time if the child misbehaves (*n* = 329)	239	72.6		
Child has access to game consoles (*n* = 330)	81	24.5		
Allow meals to be eaten in front of TV (*n =* 330)				
Frequently	173	52.4		
Rarely/never	157	47.6		
Allow snacks to be eaten in front of TV (*n* = 329)				
Frequently	260	79		
Rarely/never	69	21		
Allow children to play outside actively (*n* = 332)				
All the time	111	34.7		
Not all the time	221	65.3		
Allow children to play inside actively (*n =* 330)				
All the time	212	64.2		
Not all the time	118	35.8		
Home physical environment				
TV on whether or not it is being watched (*n* = 332)				
Most of the time/All the time	70	21.1		
Sometimes	121	36.4		
Rarely/never	141	42.5		
Presence of outdoor garden and space (*n* = 328)	300	91		
Usable play equipment (*n* = 326)	161	48.5		
Number of televisions in the home (*n* = 332)			1.0	1.0–2.0
TV in a room where meals are eaten (*n* = 331)	137	41.4		
TV available in child’s bedroom (*n =* 330)	44	13.3		
Game console available in child’s bedroom (*n =* 330)	19	5.7		
Cable or satellite TV (*n =* 330)	281	85.2		

IQR: interquartile range; TV: television.

**Table 3 ijerph-18-01589-t003:** Participant characteristics and home environmental characteristics (parents’ rules and practice and home physical environment) associated with children’s TV screen time after bivariate analysis.

Participant Characteristics	Children’s TV Screen Time				
	<1 h Daily		≥1 h Daily		
	*n*	%	*n*	%	*p* ^1^
Children					
Attending pre-school					
Full-time	69	47.9	51	27.4	<0.001 ^2^
Part-time	75	52.1	135	72.6	
Parents					
Education level					
Undergraduate and post-graduate education level	98	67.1	90	48.4	0.001 ^2^
Secondary school or less education level	48	32.9	96	51.6	
Parents’ sedentary behavior (range 2 to 12)	median	IQR	median	IQR	*p* ^1^
Watching TV	3.0	2.5–3.5	3.0	3.0–4.0	<0.001 ^3^
Sitting doing paperwork or computer work	3.0	2.0–3.5	2	1.0–3.5	<0.001 ^3^
Parents’ rules and practice around children’s TV screen time	*n*	%	*n*	%	*p* ^1^
Set rules around TV and game console					
Yes	123	87.8	140	75.7	0.040 ^2^
No	22	15.2	45	24.3	
Child has access to game consoles					
Yes	22	15.1	59	31.9	<0.001 ^2^
No	124	84.9	129	68.1	
Allow meals to be eaten in front of TV					
Frequently	54	37.2	119	64.3	<0.001 ^2^
Rarely/never	91	68.2	66	35.7	
Allow snacks to be eaten in front of TV					
Frequently	92	63.9	168	90.8	<0.001 ^2^
Rarely/never	52	36.1	17	9.2	
Allow children to play outside actively					
All the time	62	44.0	50	27.5	0.002 ^2^
Not all the time	79	56.4	132	72.5	
Home physical environment					
TV on whether or not it is being watched					
Most of the time/All the time	93	63.7	48	25.8	<0.001 ^2^
Sometimes	45	30.8	76	40.9	
Rarely/never	8	5.5	62	33.3	
TV in a room where meals are eaten					
Yes	41	28.1	96	51.9	<0.001 ^2^
No	105	71.9	89	48.1	
Cable or satellite TV					
Yes	113	77.4	168	91.3	<0.001 ^2^
No	33	22.6	16	8.7	
	median	IQR	median	IQR	
Number of televisions in the home	1.0	1.0–2.0	2.0	1.0–2.0	0.014 ^3^

IQR: interquartile range; TV: television. ^1^
*p* < 0.05 was significant; ^2^ Association between categorical variables assessed using the chi-squared test; ^3^ Association between non-normally distributed continuous data assessed using a Mann–Whitney U test.

**Table 4 ijerph-18-01589-t004:** Association between participant characteristics, home environment and children’s TV screen time.

Participant Characteristics	Children’s TV Screen Time		
	Adjusted ^1^		
	OR	95%CI	*p ^2^*
Children			
Attending pre-school			
Full-time ^(Ref.)^			
Part-time	1.18	1.03–3.18	0.039
Parents			
Education level			
Undergraduate and post- graduate education level ^(Ref.)^			
Secondary school or less education level	1.27	0.59–2.74	0.530
Parents’ sedentary behaviors			
Watching TV	1.65	1.09–2.50	0.018
Sitting doing paperwork or computer work	0.80	0.60–1.07	0.136
Set rules around TV and game consoles			
Yes	1.37	0.63–2.96	0.424
No ^(Ref.)^			
Child has access to game consoles			
Yes	1.51	0.65–3.49	0.331
No ^(Ref.)^			
Allow meals to be eaten in front of TV			
Frequently	1.07	0.50–2.01	0.984
Rarely/never ^(Ref.)^			
Allow snacks to be eaten in front of TV			
Frequently	2.66	1.17–6.06	0.019
Rarely/never ^(Ref.)^			
Allow children to play outside actively			
All the time ^(Ref.)^			
Not all the time	2.01	1.04–3.88	0.036
Home physical environment			
TV on whether or not it was being watched			
Most of the time/All the time	3.38	1.10–10.38	0.032
Sometimes ^(Ref.)^			
Rarely/never	0.35	0.18–0.71	0.004
TV in a room where meals are eaten			
Yes	2.35	1.27–4.34	0.006
No ^(Ref.)^			
Cable or satellite TV			
Yes	1.01	0.40–2.55	0.982
No ^(Ref.)^			
Number of TV in the home	0.98	0.66–1.44	0.926

CI: confidence interval; OR: odds ratio; TV: television; ^1^ Values are OR that were obtained from the final binary logistic regression model, statistically significant (x^2^ 14, *n* 317) = 112.333, *p* < 0.001, model explained 40% (Nagelkerke R^2^) of the variance in children’s TV screen time and correctly predicted 76% of cases. ^2^
*p* < 0.05 is significant.

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
