# Peer review of "Associations between Home Environment, Children’s and Parents’ Characteristics and Children’s TV Screen Time Behavior"

_ijerph, 2021, doi:10.3390/ijerph18041589_

Round 1

Reviewer 1 Report

This manuscript describes an interesting study exploring parent characteristics and behaviours and the associations with young children's screen time. It is a good study that conveys useful information that is timely. The majority of my comments are around the statistical approach and the rationale thereof. I hope the authors find this feedback helpful. 

Reviewer 2 Report

GENERAL COMMENTS

The manuscript aimed was declared to investigate any possible differences in association among reported parental role modelling, rules and practices and 4 to 5 years kids’s TV screen time engagement.

Several concerns emerge while reading the manuscript about the sampling, about regarding the questionnaire in use and variable choices, as a consequence data analysis and results presentation and interpretation.

On the overall the manuscript has the characteristics for being an extensive research report rather than a specific and scientific investigation. Stated the study aims, authors appear to address all the available variables intuitively associated with the main scope of the study, without a precise and well documented question. In particular they intend to test parental role modelling, rules and practices on the respect of TV viewing indications for toddlers and preschoolers..

Major issues:

Introduction

Given the study aim, the introduction is generalist and superficial. For examples authors should better explore the main research focus of what is intended for parental role modelling and sounding theories beyond the investigated relationships.

Because if these shortcomings, the approach then followed is that of a collection of interesting facts and numbers that show to be differently associated with the study outcome losing its initial intent. Unfortunately, this choice takes the reader (and the authors) into a plethora of numbers, proportions and possible associations without the theoretical or conceptual framework needed to allow a robust reasoning.

Line 49 page 2: the reasoning seems to open on role modeling, while it fails. Moreover, it is intuitive but not clear the association authors are reporting along with the cited literature.

Materials and methods

It is not sufficient to state “Details of the participants’ recruitment, inclusion/exclusion criteria and data collection procedures are described elsewhere”. Readers can be forced to access and download a further manuscript to judge whether study characteristics, study power, sampling procedures, rate on response, and the overall methodology are to be considered suitable enough to proceed on.

It is strongly recommended improve this section.

Several concerns on both the study internal and external validity.

Point 2.1 line 68

Sampling procedures and more in general the catchment area of inquiry: The study has the ambition to be representative of the Irish population (line 61 and 284/285). Given that the Irish population is nearly about 5 millions of inhabitants over an area of 68 890 Km2, of about the 8% in the 0 to 4 age category. Is it a sample of 332 children big enough to allow any inference on the outcomes under study?

Unit of sampling and unit of analysis: what is the sampling unit? Schools, families, kids? where the kids sampled within school of within the community? Can the choice, whatever it is, have any effect on the possibility to provide any inference on the Irish population?

Point 2.2.1 line 74

What about non response rate? No information is given.

Interviewees: were both parents invited to replay to the questionnaire? As results reports a higher proportion of females, it is like to suspect a self-selection of participants.

Questionnaire in use: no details on the questionnaire, answering procedures (e.g. answer were collected by means of an interviewer or it just was a self-compiled tool, was it paper of electronic, were the participant volunteers or subjected to a personal reward for participation).

Sources declared were the “Pre-Schoolers Health Study” for part of the questions and “Healthy home Survey” for another part. No assessment on the internal validity no any psychometric study was performed to assess item stability. Questions are formulated to ask for an “average” of time spent in a “typical” week in each of the activities inspected. Avoiding to force the participants to focus on a specific moment in the past, the study should discuss as a limit the reduced comparability on the collected answers. In particular the study aim is weather and seasons dependent (time spent outside in winter is intuitively less than in spring or summer) deserving a higher precision.

Point 2.2.2: again, the question here asks “on average” leaving to much room for personal adjustments. This issue was extensively discussed in the dedicated literature, as a consequence it has to be included among the study limits. Moreover, as DVD and videos are nowadays relatively less common than web videos, the question should have been introduced with a short explanatory text, to guide respondents to eventually consider all media available. Unfortunately, the manuscript is rather poor in its methodological issues, and because of this it is strongly recommended to revise and enlarge the whole section to be more informative.

Just to give further examples, Line 113 p 3 Scale adjustment: it is not immediately clear the choice on used cut points used and justifications for it. No investigation on the presence of brothers, as it can likely act as a moderating factor for the outcome in analysis.

Point 2.2.3. it is not clear whether aswers about parents’ behaviours refer to the respondents (in this case mother) or on the family on the whole (here comprising the father behaviours). Given the suspect of self-selection which seems to emerge in the study and given that inspected behaviours have been described to be not equally distributed between parents, and further complicated in atypical family structures (not investigated), this issue is particular important and since it is exactly the manuscript aim, it deserves lot more attention.

Results section, as already anticipated, remembers that of a report. Too much numbers and variable, part of which are theoretically distant from the broader study scope. I would suggest the authors to reframe the whole manuscript by choosing a limited number of variables, preferably coherent with stated aims and well supported in the introduction.

Discussion needs to be deeply revised to be adapted to the new manuscript design. As a suggestion we invite the authors to check for explanatory statement non adequately supported.

Finally line 284 the authors declare their study to be the first to inspect the studied association in Ireland. As Ireland gave the birth to the “Growing Up longitudinal study”, I suggest to be more cautious.

Reviewer 3 Report

It was a pleasure to review this manuscript.  This study will make an important contribution to our understanding of behaviours in the home that predict children's screen time.  As the authors have noted, excessive screen time is a major issue in most countries.  Also noted are the increasing number of national guidelines on screen time as countries try to advise parents on managing screen time and the problems of prolonged screen time.  The findings from the present study provide evidence for strategies such as allowing more time for outdoor play.  It is interesting that children of parents doing paper/computer work were less likely to watch more than an hour of TV per day - perhaps this is associated with education and income?  Something to explore in future research, but it is interesting that it seems to be parental behaviours related to tv screens rather than other screens that are the main predictors.

I have a minor suggestion for a revision, but it is optional and is more of a citation/writing issue than related to the quality of the study.  Line 247 of page 11 has citations to 4,27-29 (i.e. reference to other studies), but the sentence (lines 245-247) seems to be about behaviour of parents within the present study and therefore no citation is needed.  If there is a citation, it could be to indicate consistency of present findings with previous studies.

I appreciated the clarity of the article.  The introduction was clear and concise.  The measures were clearly described.  Care was taken to lead the reader through the analysis - all decisions were transparent.  The discussion supported interpretation of the results and also emphasised the limitations of the study.  The conclusions were very helpful for translating this research into policy/practice.

Reviewer 4 Report

Associations between parental role modeling practices, home physical environment and pre-school children’s TV screen time behavior

This paper explores associations between parenting practices, home environment and screen time behaviour in pre-school children. The paper is clear and well-written. I recommend the following edits to the manuscript:

Line 36: The reference is 2011 for a behaviour that will have significantly changed over the past 9 years, please can you check there is not more recent literature regarding what is the most common sedentary behaviour among under 5’s.

Line 43: Whilst this paper was submitted before the new WHO guidelines were published on sedentary behaviour, please check the information is still up to date and update the reference.

Line 70: Whilst details are given elsewhere, a short overview included in this paper would be very helpful as a reader should not need to go to another paper for the basic information.

Line 87: “Any type of TV” does that include if the children were watching the videos on an iPad or was it specifically on a TV device?

Line 97: Please change < 5 days to 1-4 days, as you have a none category, which makes < 5 unclear. Please change in the results section as well.

Line 142: Where you have ‘over 60%’ and then you present the number. Please re-write the sentence to integrate the 63% into the sentence to reduce repetition.

Line 151: The label “working” has been used for sitting doing paperwork or computer work. Please use the sitting doing paperwork/computer work in the text to accurately reflect the question asked, as working is much broader than just paperwork/computers.

Line 154: the start of the sentence says “the majority of parents reported…” and then says 42.5% rarely left the TV on. This is not the majority. Please correct the sentence.

Table 3 – the adjusted footnote is missing

Please add the non-significant adjusted data to the tables, whilst the data may remain non-significant it is still important to see. 

On line 169 the authors state the comparison between full-time and part-time nursery attendance. I would like the authors to discuss this in the discussion highlighting caution in its interpretation as parents are unable to comment about the time in nursery which could alter the effect.

Round 2

Reviewer 1 Report

I wish to commend the authors for addressing the comments as they did - The paper reads very well. 

Author Response

We appreciate this positive comment. Thank you very much, your comments and feedback really helped strengthen and improve the quality of the re-submitted manuscript. 

Reviewer 2 Report

The manuscript was extensively rewritten and many of the suggestion were someway managed. Unfortunately, from this side,  several doubts remains. 

One above all, the non respondence rate was really hight, to hight to be completely ignored. Non-response becomes a really critical issue when response rates fall below 70%, and here it seems to be assessed at less than 50%.  Significant association between variable in study and non-response are alone a matter of study invalidation. 

Even if suggests authors seem to have ignored the issues, allowing the reader to seriously raising a wall of distrust on the whole work.

As a reviewer I appreciate the efforts in having adjusted the manuscript, and at the same time I appreciate the trust given by the other reviewers. Unfortunately I don't feel comfortable enough in allowing the study to be published
